# Operator Identification in a VR-Based Robot Teleoperation Scenario Using Head, Hands, and Eyes Movement Data

## ABSTRACT

Remote teleoperation using a Virtual Reality (VR) allows users to experience better degrees of immersion and embodiment. Equipped with a variety of sensors, VR headsets have the potential to offer automatic adaptation to users' personal preferences and modes of operation. However, to achieve this goal VR users must be uniquely identifiable. In this paper, we investigate the possibility of identifying VR users teleoperating a simulated robotic arm, by their forms of interaction with the VR environment. In particular, in addition to standard head and eye data, our framework uses hand tracking data provided by a Leap Motion hand-tracking sensor. Our first set of experiments shows that it is possible to identify users with an accuracy close to 100% by aggregating the sessions data and training/testing with a 70/30 split approach. Last, our second set of experiments show that, even by training and testing on separated sessions, it is still possible to identify users with a satisfactory accuracy of 89,23%.

**ACM Reference Format:**
Anonymous Author(s). 2023. Operator Identification in a VR-Based Robot Teleoperation Scenario Using Head, Hands, and Eyes Movement Data. In *Proceedings of 6th International Workshop on Virtual, Augmented, and Mixed Reality for Human-Robot Interaction (VAM-HRI) (VAM-HRI 2023)*. ACM, New York, NY, USA, 9 pages. https://doi.org/XXXXXXX.XXXXXXX

## 1 INTRODUCTION

Teleoperation using various Virtual Reality (VR)-based interfaces has been demonstrated to be more efficient compared to standard PC-based controls and allows users to experience a better degree of immersion and embodiment [11]. The improvements are observed in objective task performance measurements as well as subjective quality of operation data from the users.

The main benefit of VR-based interface, which is its ability to utilise a larger extent of the operator's movements, comes with certain risks in terms of users' privacy. Because of the increased observability of the user, VR-based interfaces possess greater security concerns than other traditional human-machine interfaces. For example, movement data readily available even in consumer-grade VR equipment is powerful and, if not properly protected, can be used in ways that violate user integrity and security [5].

The main goal of this paper is to investigate the feasibility of identifying a user within a group of participants using their VR

*VAM-HRI 2023, March 13, 2023, Stockholm, SE*
© 2023 Association for Computing Machinery.
ACM ISBN 978-1-4503-XXXX-X/18/06…$15.00
https://doi.org/XXXXXXX.XXXXXXX

behavioural data, gathered while performing a teleoperation task. Previous work on user identification in VR focused on dynamic situations, such as ball-throwing tasks [2, 8]. This is problematic, as those tasks exhibit characteristics that can be found in entertainment scenarios, but are not easily transferable to teleoperation situations where operators stand semi-still throughout their duties. In fact, while the literature shows that head position and orientation play a critical role in identifying a user playing on a VR game, in this paper we will show that this might not be the case for VR teleoperation scenarios. In this study, we implemented a scenario that require the users to teleoperate a robotic arm and move a set of coloured cubes, according to our instructions. These tasks involve specific user movements and behaviour that recall the ones typically seen in teleoperation case studies. The contributions of this paper are the following:

(1) An experimental VR framework that allows users to teleoperate a robotic arm in a realistic industrial environment;
(2) A teleoperating scenario that involves realistic operator movements and behaviour, in general;
(3) A data collection pipeline to collect users' behavioural data during VR teleoperation sessions;
(4) An in-depth evaluation of accuracy results yield using different combinations of Machine Learning (ML) classification algorithms and datasets.

## Outline

The paper is organised as follows: Section 2 gives an overview of previous work related to VR user identification, cybersecurity in VR environments, and use of the Leap Motion device for identification purposes. Section 3 describes the implemented system from hardware setup to model analysis, and Section 4 describes the experimental setup that we designed. Section 5 presents the results obtained in terms of identification accuracy. Finally, in Section 6 we discuss the limitations that concern our work, and in Section 7 we draw our final conclusions and reflections.

## 2 RELATED WORK

Several works have previously investigated user identification using motion data. For instance, Liebers et al. [9] achieved a 90% identification accuracy with a deep-learning classifier, using the spatial motion data of 16 participants gathered on a set of bowling and archery VR tasks. Kupin et al. [8] proved that it is possible to identify users that undertake a ball-throwing VR task by simply analysing the trajectory of their dominant hand (i.e., using the controller position feature). With their approach, the authors obtained an overall accuracy of 90% over 135 3D trajectory points, and an accuracy of 92.86% with 95 points. Kupin et al. [8] improved the results of Ajit et al. [2], achieving a best highest accuracy of 93,03%. Their result was obtained by adding the recessive hand and head

position to the analysed features, as well as hands and head orientation features. Their results suggest that hand and head orientation are stronger identification features than position information.

Mathis et al. [13] proposed a combination of knowledge-based authentication and biometric authentication to achieve double-layered security in VR. Their study shows that a fully convolutional network (FCN) over a combination of dominant and non-dominant hand position and rotation produces the most accurate results, with an accuracy of 98.91%. Following the intuition that human bodies are all slightly different from each other, Pfeuffer et al. [17] used spatial relations between body parts as features for identification, showing that the distance between headset and controllers can be used as an additional identification feature.

Most modern VR headsets are equipped with eye-tracking technology. Various papers have shown that high accuracy results can be obtained using eye data for identifying users. Rogers et al. [18] obtained a 94% accuracy using as features blink and head movement data of users exposed to changing images. Liebers et al. [10] also used a combination of head and eye movements, obtaining a 75% accuracy with a kNN (k=1) algorithm and 100% accuracy with a deep-learning approach. Olade et al. [15] combined gaze with hand movement and achieved an authentication system with a 98.6% accuracy that, in addition, proved to be reasonably resistant to impersonation attacks. Last, Luo et al. [12] used electrodes to collect electrooculography (EOG) signals and showed that eye globe movement and eyelid movement can also be used for identification/authentication, achieving EERs as low as 4.97% against statistical attacks and 3.55% against impersonation attacks.

The works mentioned so far show the possibility of identifying users using their VR-related information. To prevent misuse, clear policies about data collection and data usage should be in place, but in 2018 Adams et al. [1] showed that this is not the case. The authors analysed the VR experiences that can be downloaded for Oculus and HTC, highlighting that 82% (74) and 30% (15), respectively, provided a privacy policy. However, among these Oculus experiences, only 19% mention VR-specific data collection, while 33% of the 15 HTC experiences discuss the use and collection of VR data.

Beyond misuses of data by manufacturers and developers, external malicious attackers can compromise VR setups and de-anonymise users against their will. ReAvatar [6] is a deanonymisation attack that allows third parties to identify users using their avatar positional data. The attack shows high accuracy in identifying users among a small group of participants (N = 5), despite using different avatars over different sessions. Arafat et al. [3] proposed a method to detect virtual keystrokes from users by extracting their hand movement data from WiFi network signals. These attack methods show that VR interaction data could compromise users' privacy even when privacy policies are in place, as it can be stolen by unintended parties.

A popular category of attacks on VR authentication techniques is the over-the-shoulder category, which consists of an external attacker observing a VR user, trying to decode the movement patterns used to authenticate on the system. Various papers take into account these attacks and try to address them directly with different combinations of knowledge-driven and biometrics-driven approaches [14][13][7]. However, the system must store legitimate authentication data to enable the authentication process: if stolen, the authentication template could be used by malicious third parties [20]. Therefore, a VR system that uses behavioural templates should take into account the risk of theft and implement protection strategies, such as the canceling biometric technique [16].

Most studies use the standard controllers included in consumer-grade VR systems. In our research, we investigate the integration of the Leap Motion hand-tracking sensor with standard VR sensors to identify users. Although, to the best of our knowledge, our work is the first attempt in this direction, other papers showed that the Leap Motion sensor holds promise in this regard. Atas [4] showed that hand tremors, detected using the Leap Motion hand X-axis, allow high-accuracy identification of users. Xiao et al. [21] captured users' Leap Motion behavioural data to implement a biometric authentication method, which achieved an average EER of 3,8%.

Last, Sugrim et al. [19] noted that, while the results of many studies are promising, most of them rely on a sample size too small for drawing reliable conclusions, since that accuracy tends to deteriorate with sample sizes larger than 20 participants. In fact, of the 30 articles surveyed by Sugrim et al., 77% relied on a pool of less than 20 participants.

Among the sources (N=32) that were reviewed in preparation for this study, a total of 15 explicitly handled the topic of biometric identification/authentication. Out of these 15 sources, 9 (60%) made use of hand related biometrics using controllers, 12 (80%) made use of head related biometrics, 5 (33%) made use of eye related biometrics, and 2 (13%) made use of hand tracking biometrics using Leap Motion hand tracking although not performed in a VR setting. As the use of Leap Motion hand data for identification in VR is a novel concept, it is one of the main points of interest when designing the data collection and VR experiment environment.

## 3 ARCHITECTURE AND IMPLEMENTATION

In this paper, we propose a system architecture for teleoperating a robotic arm on VR and collecting user behavioural data, shown in Figure 1. For the hardware part, we mounted a Leap Motion hand tracking sensor on a HTC Vive Pro Eye headset. For the VR scenario, we designed a scene for robot teleoperation in Unity 3D game engine, which also handles the VR rendering part. For safety reasons, instead of operating the real robotic arm, we simulate the arm and the entire physical environment in CoppeliaSim[1] simulator, using Billet v.2.82 physics engine. Unity and Coppeliasim interface to each other through the ROS[2] ecosystem, using ROSBridgeServer and a corresponding Unity client. This approach allows for switching seamlessly between the simulated instance and the real robot. Last, we capture gaze data via SRanipal API, hand data from LeapMotion sensor via the LeapMotion SDK, and we stream-record all data into log files.

For this study, we used a HTC Vive Pro Eye[3], alongside Viveport v1.4.15.3, to handle software updates and setup the HTC Vive Pro Eye kit. As a hand-tracking sensor, we used the Leap Motion Controller[4] by Ultraleap, which is an optical hand tracking module for capturing hand movements. The Leap motion sensor is mounted on the front of the HTC Vive headset, as shown in Figure 2.

---

[1]https://www.coppeliarobotics.com/helpFiles/en/dynamicsModule.htm
[2]https://www.ros.org/about-ros/
[3]https://www.vive.com/us/product/vive-pro-eye/overview/
[4]https://www.ultraleap.com/product/leap-motion-controller/

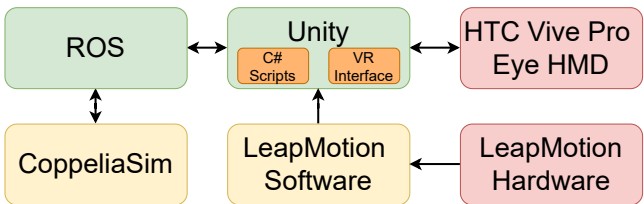

**Figure 1: System diagram of the experimental setup. The teleoperation link is established between Unity and ROS (in green). CoppeliaSim is used as a task space simulator. LeapMotion and HTC helmet (in red) are connected to Unity via corresponding SDKs.**

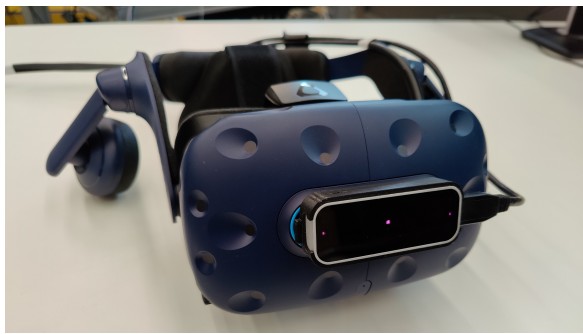

**Figure 2: The HTC Vive Pro Eye VR headset used in the study, with the Leap Motion hand tracking sensor mounted on the front. The setup allows for continuous hand tracking as the user operates in the VR space.**

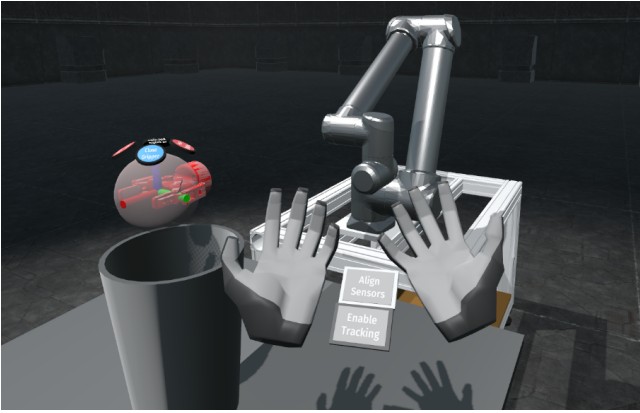

**Figure 3: A user view of the experimental scene with both hands tracked. The robot is in "parked" position, the interaction proxy (semi-opaque ball) is shown over the bucket.**

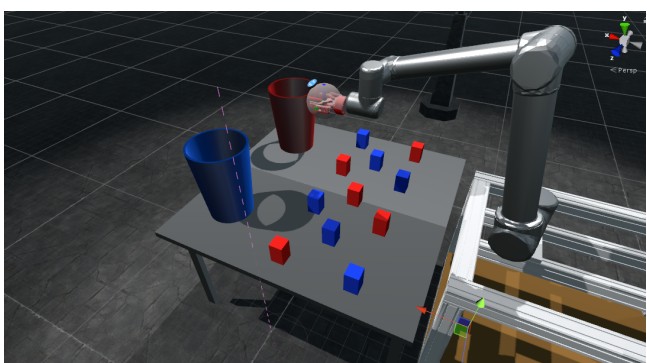

**Figure 4: The task space as seen in the VR teleoperation interface. The robot is in engaged mode, tracking the pose of the interaction proxy. The participant is instructed to use the robotic arm to sort the blue cubes into the blue bucket and the red cubes into the red bucket.**

The user interface used for the data collection was developed using Unity3D[5]. The interface allows users to interact in real time with the experimental environment. The Unity scene contains a virtual instance of the robotic arm, identical to the physical arm remotely operated, that receives real-time data about objects in the task space. Users are able to see the simulated robotic arm, buckets, and cube objects, and interact with the task space using an interaction proxy, as seen in Figure 3. The user point of view of the VR interface is shown in Figure 4, and the corresponding simulated scene is shown in Figure 5.

CoppeliaSim is a robotics simulator with integrated development environment that is used to simulate the robotic arm and the objects in the scene. Figure 5 shows the task-driven scene as implemented in CoppeliaSim. For this project, we picked the physics engine bullet[6].

The simulated robotic arm is a model of a UR10[7] robotic arm with a gripper attached. The gripper is a Robotiq 85[8] gripper. The robot tracks the pose of the interaction proxy, sent from the interface part, using damped-least-squares (DLS) inverse kinematics (IK) solver. In fact, two independent IK groups are used for the robot: one tracking position of the proxy and another to track orientation. The latter

has higher damping factor to compensate for possible sporadic rotational movements of the proxy.

The LeapMotion hand tracking sensor uses the Leap Service Provider, included in the assets package, to communicate with Unity. The Service Provider works by providing frames and images from the hand tracking sensor to other parts of the application. Due to the frame rate of the Unity interface running at 90Hz, the corresponding Leap Frame data is gathered at the same frequency of 90Hz.

By default, when one of the participant's hands is not detected by the Leap Motion sensor, hence not tracked, the Leap Motion software outputs the last known positional data for that hand. This could create uncertain data interpretation, as the same information would represent the detection of a hand completely still. To avoid this potential source of confusion, when a hand is not detected, the data collection script stores vectors and quaternions of zeros (0.0, 0.0, 0.0), instead of the last known information.

---

[5]https://unity.com/
[6]https://pybullet.org/wordpress/
[7]https://www.universal-robots.com/se/produkter/ur10-robot/
[8]https://robotiq.com/products/2f85-140-adaptive-robot-gripper

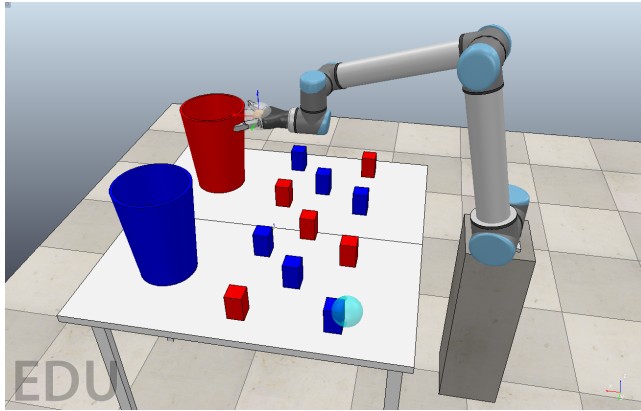

**Figure 5: The task space as seen in the simulation environment in CoppeliaSim. All interaction physics of the robotic arm, buckets, and cubes are simulated in Coppelia and streamed into the Unity interface.**

This approach allows to take into account participants' different habits of dealing with fatigue (e.g., dropping their hands to the sides or otherwise move them out of tracking range ) as a uniquely distinguishable feature. The data that we collected from the Leap Motion, stored in different data files for each hand, contains the following:

- Position of each palm, represented by a vector of floats (x,y,z);
- Rotation of each palm, represented by a quaternion of floats (x,y,z,w);
- The individual position of each fingertip, represented as a vector of floats for each finger (x,y,z);
- The individual direction ("pointing direction") of each fingertip, represented as a vector of floats for each finger (x,y,z).

The hand tracking data is split into 12 data files for each hand. This modular approach makes feature selection in later stages highly modifiable, as it allows to choose individual hand data files, without the need for splitting or parsing files.

The HTC Vive Pro Eye provides position and rotation of the head, as well as eye-related data. The eye-tracking data includes pupil diameter, eye gaze direction, and eye gaze origin. The data is provided both as individual sets of data for each eye, as well as a combined average of both eyes. The total data that we collected is:

- Gaze origin of each eye, and average of both eyes, represented by a vector of floats (x,y,z);
- Gaze direction of each eye, and average of both eyes, represented by a vector of floats (x,y,z);
- Gaze origin of each eye, and average of both eyes, represented by a vector of floats (x,y,z);
- Pupil diameter of each eye, and average of both eyes, represented in millimetres by a single float;
- Head position, represented by a vector of floats (x,y,z);
- Head rotation, represented by a quaternion of floats (x,y,z,w).

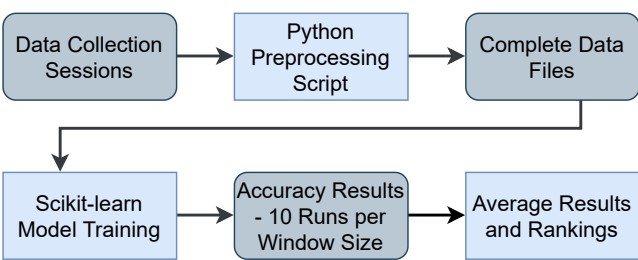

**Figure 6: Complete processing pipeline for the study. Collected data is pre-processed to select features of interest, build mean and variance data using different window sizes, and structure the data. The dataset is then split and used for training and testing on different classification models.**

## 4 EXPERIMENTS

As we show in Figure 6, the first stage consists of data collection, followed by data pre-processing. The third step is training of different machine learning models, and the pipeline ends with the evaluation of the models.

### 4.1 Data Collection

We recruited 10 participants among the faculty and student body at our university. In accordance with the university data collection policies and the national data protection law GDPR, participants signed a data release form before being enrolled. The consent form is provided in both English and the national official language, to ensure that all participants are informed about their rights concerning their personal data.

After signing the consent form, each participant fills out a questionnaire with basic information such as age, gender, and dominant hand.Then, we assign them a random 3-digit number, used as an ID label for their data and questionnaire. This is necessary to anonymise the data and, at the same time, allow us to comply with the data protection regulations. In case a participant decides to withdraw their consent, we must be able to retrieve their data for deleting the related information.

After the participants fill out the documents, we proceed to introduce the experiments. We explain the objective of the study, how to interact with the robot through the interaction proxy, and we provide some information about the limitations of the robotic arm and the VR environment. Finally, we calibrate the VR headset eye-tracking and the participant can start the experiment.

To collect the data, we designed two different types of sessions. The first is a *free play* session, in which the participants are free to play around as they prefer. The second is the *task driven* session, where the participants are instructed to complete a specific task, adhering to a specific set of instructions that we describe later in this section. Each participant of the study performs one free play session and two task driven sessions, in a casual order. Some participants start with the free play, followed by the two task-driven scenes, while others do the task-driven ones first, and the free play last. Each participant has the option to take a break between sessions. To collect enough data from every session and preventing participants

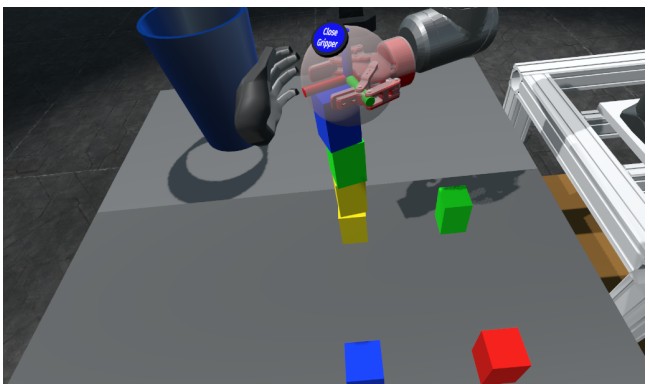

**Figure 7: A participant stacking blocks in a free play data collection session. The free play sessions produce user motion data that differs from the task driven sessions as the user has no constraints.**

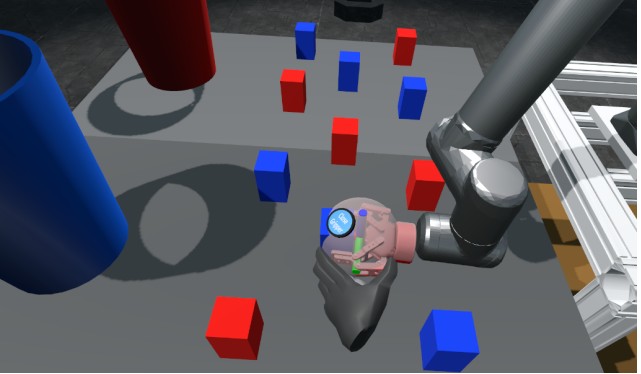

**Figure 8: A participant attempting to pick up a block while avoiding other blocks. Proper rotation and positioning of the robotic arm is required.**

from interrupting their free play session too early, we set a goal duration of 5 minutes for each free play session.

For what it concerns the free play scene, as shown in Figure 7, we created a scene with random coloured cubes and a bucket at the top right corner of the table. During the briefing before the session, we inform the participants that they can do what they prefer for 5 minutes. We also provide a couple of ideas about possible actions, such as stacking blocks and dropping the blocks in the bucket.

The task-driven scene is designed to be a slight challenge for the participants. We placed 11 blocks and 2 buckets in the scene, coloured red and blue as seen in Figure 8. Then, we instruct the participants to pick up the blocks and sort them into the buckets, according to their colour. In one of the two task-driven sessions, the participants have to sort the blue cubes in the blue bucket followed by the red cubes. For the other task-driven session, the order is reversed. For both sessions, the participants have to try to avoid knocking over blocks and ignore unreachable blocks (e.g., blocks that fell off the table or got stuck). The instructions are meant to encourage the participants to be careful and deliberate when operating the robotic arm, which helps to extend the completion time of most sessions beyond the 5 minutes mark.

## 4.2 Data pre-processing

Before we can input the collected data into machine learning algorithms for training, it is necessary to pre-process the information. Our data collection script collects the raw information of each participant session in a single folder, split into sub-folders for head, left/right hand, and eye data. The session folders are then renamed using a naming convention, which is integral to the pre-processing. The naming convention contains participant id, session type, and session number, as follows:

$$P\_PARTICIPANT\_SESSIONTYPE\_S\_SESSIONNUMBER.$$

As an example, the following name represents a session for a participant with id number 1 who performed a task-driven session, which is also the participant's second session:

$$P\_1\_T\_S\_2.$$

As mentioned, the naming convention is an integral part of the preprocessing phase, meaning that we developed a Python script that makes use of the folder hierarchy and folder names to sort, combine, and label the data in the CSV files.

One problem encountered is that different CSV files have columns with overlapping names (such as, X, Y, and Z). To solve the confusion that would arise from having multiple columns with the same name, the script iterates through each CSV file and renames the columns by adding the name of the body part, as well as which side (left or right) it belongs to. After renaming, each column in each session is uniquely named, avoiding naming overlaps from occurring when combining data. An illustration of the parsing process can be seen in Figure 9.

Completed the renaming, we combine horizontally each session in Pandas[9] data frames, we calculate the mean and standard deviation using a rolling window of varying sizes. Then, we store in CSV files, one per each experiment session, the resulting data frames with mean and standard deviation for the rolling windows. The window sizes are $[5, 10, 20, 40]$ *seconds*, which equal to $[450, 900, 1800, 3600]$ frames, respectively. All session CSV files are combined vertically in one data frame for each participant and stored in new CSV "meta" files (again, one per participant). Finally, all meta files are combined in a single CSV file, containing the final and complete set of mean and standard deviation data, labelled.

## 4.3 Machine Learning and Models Training

In this paper, we intend to answer two questions. First, if it is possible to identify a user by training the model on a session type (e.g., the free session) and testing it on a different session type (e.g., the task-driven). If the answer is positive, we want to find if the accuracy is comparable to the accuracy obtained by training and testing the models on a combined dataset with a classic 70/30 split. Our hypothesis is that it is possible to do so, but that learning and testing on separated datasets results in decreased accuracy with respect to combining the datasets and adopting a 70/30 split.

Starting from the raw data, organised in CSV files as discussed, we constructed a total of six datasets that contain different portions

---

[9]https://pandas.pydata.org/docs/index.html

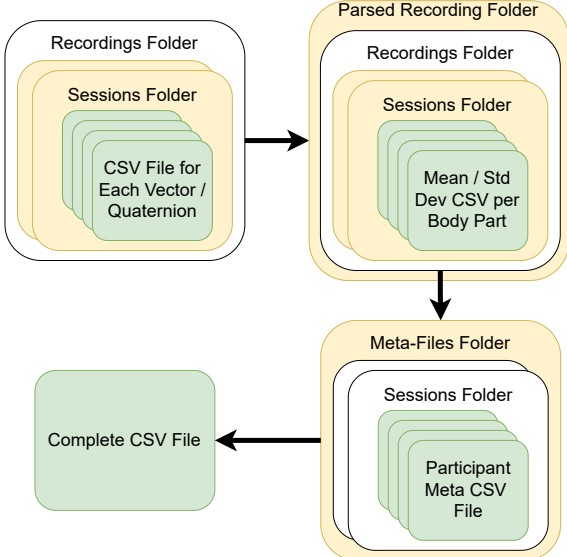

**Figure 9: Illustration of the data file parsing process. The script combines the data files and computes the mean and standard deviation of each feature.**

**Table 1: Average performance rating across all data sets. Each algorithm was assigned an accuracy ranking for each data set, and the average ranking was computed.**

| Algorithm | Average Rating | Performance placing |
|---|---|---|
| Linear Discriminant Analysis | 2.00 | **1** |
| SVM | 2.67 | **2** |
| Logistic Regression | 3.17 | **3** |
| Random Forest | 3.67 | **4** |
| kNN (N=5) | 4.50 | 5 |
| kNN (N=10) | 5.00 | 6 |
| kNN (N=1) | 5.67 | 7 |
| Decision Tree | 7.50 | 8 |

of the data. In particular, we created three basic datasets with every feature of hand, head, and eye data, respectively. Then, we created two additional datasets, one combining hand and head data, and the other combining hand and eye data. The sixth and last dataset contains every feature of the three basic datasets aforementioned.

For what it concerns the ML implementation, we chose scikit-learnSK-Learn[10] as a library. Since that scikit-learn provides several ready-to-use ML classification algorithms, we decided to test 8 different algorithms and pick the 4 best performing. The first pool of 8 algorithm included Logistic Regression, Decision Trees (DTs), Random Forests (RFs), k-Nearest Neighbor (kNN) (for k=1, k=5, and k=10), Support Vector Machine (SVM), and Linear Discriminant Analysis (LDA). First, for each algorithm, we standardised the dataset features to minimise potential unexpected behaviour due to non-standardised data. The standardisation is done by applying scikit-learn StandardScaler function, which removes the mean and scales every feature to unit variance.

We train and test every algorithm on the entire dataset, with a 70/30 split, for 10 evaluation runs and we compute the average accuracy of the 10 runs. For each of the 10 evaluations, we rank the algorithms based on their performance. Finally, we compute the average of the ranks to obtain a final performance ranking, and we pick the best 4 performing algorithms that we further evaluate. That is, LDA, SVM, Logistic Regression, and RFs. In Table 1, we show the overall ranking.

Once we have selected the four best performing algorithms, we investigate further their performance, aiming to answer the research questions we defined earlier in this section. In particular, we evaluate the 4 algorithms over a combination of 6 different groups of features (hand, head, eye, hand/head, hand/eye, complete),

two approaches to training/testing (i.e., 70/30 split on total dataset and training/testing on separated sessions), and 4 different rolling window sizes (5, 10, 20, and 40). We perform each evaluation 10 times and compute the average accuracy, totalling up to $[4 \times 6 \times 2 \times 4 = 192]$ evaluations and $[192 \times 10 = 1920]$ runs.

## 5 RESULTS

For our investigation, between 13th and 25th May 2022, we recruited 10 participants among the faculty and student body at our university. Every participant performed three sessions each, on the same day, with optional pauses between sessions.

In this section, we analyse the results of our experiments. First, the average participant age is 25,1 years old. Data show that the group is homogeneous, with an average age of 25,1 years, heavily skewed towards participants with dominant right hand (90%), males (80%), and little to no previous experience with VR (90%). On the one hand, participants homogeneity is a problem for the generalizability of the study, as the same experiments conducted on left-handed female participants with previous VR experience might yield different results to the ones we obtained. On the other hand, with small experimental groups, homogeneity ensures that our ML algorithms do not barely discern people on evident characteristics, such as their dominant hand.

As previously covered in Section 4, we ran 192 evaluations on the data collected with our participants, 10 times per evaluation, for a total of 1920 runs. First, in Table 2, we show the result of the 96 evaluations (i.e., the average of 10 runs per each evaluation) that concern the 70/30 split approach to dataset training/testing. In the table, we mark in **bold** the best accuracy obtained across datasets and ML algorithms, given the same window size. In *italic*, the best accuracy obtained on each dataset, with varying combinations of ML algorithm and window size. In the bottom right field, we have marked the best accuracy obtained in all 96 evaluations.

Analysing Table 2, first we notice that the eye dataset performs consistently better than others: of 16 possible combinations of algorithm and window size, 12 performed their best accuracy on the eye dataset alone, followed by 3 best results on the complete dataset and one best result on the hand/eye dataset. It is also possible to notice that every dataset that includes the eye data performs well,

---

[10]https://scikit-learn.org/stable/

**Table 2: Complete accuracy results across all combinations of algorithm, rolling window size and data set for the task/free play complete dataset. Training and testing performed with a classic 70/30 split. Best score per row and column are highlighted using bold and cursive numbers, respectively.**

| Algorithm | Window (s) | Accuracy (%) | | | | | | Best Accuracy (%) for Window Size - **Bold** |
|---|---|---|---|---|---|---|---|---|
| | | Hand | Head | Eye | Hand/Head | Hand/Eye | Complete | |
| LDA | 5 | 83,98 | 46,00 | 99,18 | 89,22 | *99,67* | **99,86** | **99,86** |
| | 10 | *87,87* | 51,37 | **99,62** | *90,28* | 99,43 | 99,43 | **99,62** |
| | 20 | 85,48 | 54,33 | **99,23** | 86,44 | 97,98 | 97,50 | **99,23** |
| | 40 | 66,40 | 60,60 | **99,40** | 69,40 | 98,20 | 97,40 | **99,40** |
| SVM | 5 | 74,75 | 52,14 | 98,94 | 81,58 | 98,64 | **99,13** | **99,13** |
| | 10 | 76,40 | 57,82 | **99,15** | 78,86 | 98,53 | 98,63 | **99,15** |
| | 20 | 76,92 | 60,48 | **98,85** | 83,65 | 97,12 | 97,79 | **98,85** |
| | 40 | 80,20 | 59,60 | **98,00** | 83,40 | 96,00 | 95,60 | **98,00** |
| Logistic Regression | 5 | 76,40 | 48,42 | **99,04** | 82,99 | 98,64 | 98,96 | **99,04** |
| | 10 | 78,20 | 52,61 | **98,53** | 82,51 | 98,39 | 98,29 | **98,53** |
| | 20 | 81,15 | 58,08 | **98,65** | 87,02 | 98,08 | 98,17 | **98,65** |
| | 40 | 82,40 | 60,00 | **97,40** | 87,00 | 97,20 | 97,00 | **97,40** |
| Random Forest | 5 | 68,85 | *64,64* | **99,18** | 73,79 | 98,33 | 98,78 | **99,18** |
| | 10 | 69,53 | 60,38 | **99,05** | 71,28 | 98,06 | 97,96 | **99,05** |
| | 20 | 69,52 | 56,06 | 97,21 | 73,75 | **97,79** | 97,40 | **97,79** |
| | 40 | 73,40 | 63,40 | 96,00 | 73,20 | 97,00 | **97,20** | **97,20** |
| Best Accuracy (%) for Dataset - *Italic* | | *87,87* | *64,64* | *99,62* | *90,28* | *99,67* | *99,86* | ***99,86*** |

with results always above 96% accuracy. The head dataset is clearly the worst performer, with a best achieved accuracy of only 64,64%. This results is coherent with the VR tasks that we designed, which do not require the participants to move a lot their head. Although it does not perform as good as the eye dataset, the hand dataset provides a satisfying a best-case accuracy of 87,87%. It is worth noting that the eye dataset exhibits the largest number of features (i.e., 30 features versus 24 for the hand dataset and only 7 for the head dataset), hence the largest amount of useful information for identifying users. Overall, while every algorithm manages to reach outstanding accuracy results (with a lowest accuracy of 97,20% obtained by RFs), LDA with a rolling window of 5*s* achieves the best accuracy, which equals to 99,86%.

Our first set of evaluations show that aggregating the sessions data and training/testing with a 70/30 split approach allows for high accuracy results. However, we are interested in investigating whether it is possible to recognise teleoperating users by training and testing on different and separated sessions. In Table 3, we show the result of the remaining 96 evaluations (i.e., the average of 10 runs per each evaluation), obtained with training and testing on disjoint datasets. We apply to this table the same semantics of **bold** and *italic* that we used previously for Table 2.

As shown in Table 2, while with the complete dataset split 70/30, the average best performing dataset was the eye dataset, the situation differs when training and evaluating on separated datasets. The bold numbers highlight that, with fixed algorithm and rolling window, the complete dataset (i.e., the triplet hand, head, and eye data)

is on average the best performing one, followed by the hand/eye data with 4 cases, and the eye data alone with 3 cases. Moreover, LDA, Logistic Regression, and RFs perform the best with the complete dataset three out of four times, but the same dataset is never the best performer for SVM. In general, the eye, the hand/eye, and the complete datasets perform comparably well with accuracy results of 88,57%, 88,89%, and 89,23%. As previously seen in Table 2, the worst performing dataset is the eye dataset with a mere 39,04% and the performance of the hand dataset decreases to 73,79%. These results are consistent with the ones obtained in Table 2, showing that hand data is useful for identifying users, if combined with other meaningful data (such as eye data). Last, LDA proves to be again the best performing algorithm, with a best accuracy of 89,23% using a rolling window of 10 seconds.

## 6 LIMITATIONS

In this section, we provide an overview of the limitations that characterise our work. First, the number of participants we were able to recruit (N=10) is small and below the threshold of 20 participants beyond which, according to Sugrim et al. [19], identification results tend to degrade. Therefore, while our methodology and pipeline prove the feasibility of using hand and eye data for identifying teleoperators, larger studies would be necessary for declaring on the robustness of such a system. In our future works, we intend both to expand the participants pool and to evaluate additional machine learning algorithms.

**Table 3: Complete accuracy results across all combinations of algorithm, rolling window size and data set for the task/free play split data. Best scores per row and column are highlighted using bold and cursive numbers, respectively.**

| Algorithm | Window (s) | Hand | Head | Eye | Hand/Head | Hand/Eye | Complete | Best Accuracy (%) for Window Size - **Bold** |
|---|---|---|---|---|---|---|---|---|
| LDA | 5 | 64.88 | 31.10 | 82.61 | 63.21 | 86.12 | **87.46** | **87.46** |
| | 10 | 70.03 | 33.33 | 83.84 | 66.67 | *88.89* | *89.23* | **89.23** |
| | 20 | *73.97* | 36.99 | 83.56 | *72.60* | 86.99 | **88.36** | **88.36** |
| | 40 | 50.00 | 38.57 | ***88.57*** | 45.71 | 85.71 | 85.71 | **88.57** |
| SVM | 5 | 51.34 | 32.27 | 81.12 | 50.50 | **85.79** | 84.95 | **85.79** |
| | 10 | 48.82 | 32.66 | **86.88** | 52.19 | 86.53 | 86.20 | **86.88** |
| | 20 | 56.85 | *39.04* | 84.93 | 54.11 | **85.62** | 84.93 | **85.62** |
| | 40 | 60.00 | 31.43 | **85.71** | 52.86 | 80.00 | 80.00 | **85.71** |
| Logistic Regression | 5 | 54.18 | 32.78 | 83.95 | 54.18 | 84.28 | **84.95** | **84.95** |
| | 10 | 52.86 | 33.33 | 81.82 | 53.20 | 85.52 | **86.53** | **86.53** |
| | 20 | 58.22 | 37.67 | 78.77 | 57.53 | **86.99** | 86.30 | **86.99** |
| | 40 | 65.71 | 32.86 | 84.29 | 60.00 | 87.14 | **88.57** | **88.57** |
| Random Forest | 5 | 43.23 | 32.88 | 78.76 | 44.48 | 83.93 | **84.38** | **84.38** |
| | 10 | 44.51 | 35.32 | 81.11 | 45.72 | 84.98 | **85.49** | **85.49** |
| | 20 | 50.75 | 31.10 | 80.48 | 49.73 | **85.34** | 84.79 | **85.34** |
| | 40 | 46.29 | 34.57 | 80.57 | 49.57 | 83.86 | **84.00** | **84.00** |
| Best Accuracy (%) for Dataset - *Italic* | | *73.97* | *39.04* | *88.57* | *72.60* | *88.89* | *89.23* | ***89.23*** |

Another limitation lies in the VR scenes that we designed. On the one hand, more variety in setups and goals would allow us to collect richer and more complete information for identifying the users. On the other hand, it is worth taking into account that teleoperators manipulate specific machines (such as a robotic arm) with well-defined capabilities, in order to achieve a specific set of goals. In light of this, we deem our choices of free play and goal-oriented scenes appropriate, although limited.

## 7 CONCLUSIONS

In this paper, we proposed a framework that shows the feasibility of using head, eye, and hand data for identifying teleoperating users. In our pipeline, we tested various machine learning algorithms with two different approaches to learning/testing data splits. Combining all data and using a classic 70/30 split, we achieved a peak accuracy of 99,86%. As expected, learning and testing on separated datasets decreased the accuracy, but still allowed us to reach a peak accuracy of 89.23%. In general, hand data coupled with eye data consistently show good performance, suggesting that they should be combined together to achieve robust identification results.

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
