# OpenReview forum: "Operator Identification in a VR-Based Robot Teleoperation Scenario Using Head, Hands, and Eyes Movement Data"
_humanrobotinteraction.org/HRI/2023/Workshop/VAM-HRI — VAM-HRI 2023 Oral_

### Official Review · Program_Chairs · 2023-02-24
**Accept**

**Rating:** 7
**Confidence:** 5

**Review:**

Review 1:

Operator Identification in a VR-Based Robot Teleoperation Scenario Using Head, Hands, and Eyes Movement Data

In their work authors describe the identification process of VR robot teleoperators. Authors aggregate the results/sessions recording where they track hand movements (using Leap Tracker) and use this data to identify users.  The authors point out that is it hard to identify the person via movements since most of the tasks are usually semi-static.

I think that the strongest contribution of this article is showing how VAM technologies can be used for data collection, recording and later on for identification purposes.

Potential improvements:

I would change the main narrative of the paper. The data collection tool you developed is great and it is valuable to show some applications of what you can do with the collected data. However, I would not say that testing 4 ML models is groundbreaking. In my opinion, you should highlight the data collecting part and do more different experiments focusing on that.

That would require more participants and more tasks 🙂

Showing that this works on 15 participants is not very representative. I would also like you to see how you came up with that number. For future experiments, I would recommend increasing the number of participants.

You highlighted that one of the contributions is “An experimental VR framework that allows users to teleoperate a robotic arm in a realistic industrial environment”. However, you did not show the interaction with the real robot. I think it would be an interesting addition.

I think there are a bit unnecessary details when you talk about data preprocessing. I would remove it in the future version.

Results are really high but that can be caused by the very small dataset and simply overfitting.

Video recording would be nice.

Review 2:

This work investigates the ability of modern machine learning algorithms to identify VR-based robot teleoperators in order to tailor the VR experience to the specific user. The authors collect data regarding the users gaze, hand position and orientation, and head movement, to train four different machine learning models. They then compare different combinations of their user data and the models ability to correctly classify the users. In conclusion, the authors find they are able to correctly classify users with an accuracy ranging from 89-99%.

Strengths:
Writing: The authors provide a clear outline of their system, data collection procedures, and obtained accuracy metrics. It was easy to read and clearly explained.
Relevancy: The authors’ work is both interesting and clearly relevant to expanding the capabilities of VAM-HRI. It is currently time consuming to manually tailor and recalibrate VAM devices for every user. I look forward to seeing how the author incorporates their work into VAM devices in order to more seamlessly switch between different users.

Weaknesses:
Motivation: In the Introduction and related work, the author mentions the security risks related to being able to identify users based on data collected by VR devices. However, it is unclear how this current work helps reduce the security risks, or whether that was a goal of the paper.
Training Data: The author could have also clarified why they chose to use behavioral data to train their model. The HoloLens 2 uses biometric (iris) data, but that was not included in this study. Furthermore, I imagine an easier classification approach could combine the user's hand size, distance between the two eyes, and maximum arm distance from the head, to profile a person and classify them right away. Does behavioral data provide more accuracy?

The authors will benefit from talking about their current and future work at VAM-HRI, therefore I would argue for it to be accepted.

---

### Decision · Program_Chairs · 2023-03-02

Accept (Oral)